

# Modelling *Tradescantia fluminensis* to assess long term survival

Alex James[1], Sue M. Molloy[2,3], Agate Ponder-Sutton[1], Michael J. Plank[1], Shona L. Lamoureaux[2], Graeme W. Bourdôt[2] and Dave Kelly[3]

[1] Biomathematics Research Centre, University of Canterbury, New Zealand
[2] AgResearch, Lincoln, New Zealand
[3] School of Biological Sciences, University of Canterbury, New Zealand

## ABSTRACT

We present a simple Poisson process model for the growth of *Tradescantia fluminensis*, an invasive plant species that inhibits the regeneration of native forest remnants in New Zealand. The model was parameterised with data derived from field experiments in New Zealand and then verified with independent data. The model gave good predictions which showed that its underlying assumptions are sound. However, this simple model had less predictive power for outputs based on variance suggesting that some assumptions were lacking. Therefore, we extended the model to include higher variability between plants thereby improving its predictions. This high variance model suggests that control measures that promote node death at the base of the plant or restrict the main stem growth rate will be more effective than those that reduce the number of branching events. The extended model forms a good basis for assessing the efficacy of various forms of control of this weed, including the recently-released leaf-feeding tradescantia leaf beetle (*Neolema ogloblini*).

## INTRODUCTION

*Tradescantia fluminensis* Vell is a ground-covering perennial herb native to South America that has established in native forest remnants in New Zealand, Australia and south eastern states of the United States. In its native environment, *T. fluminensis* has many natural predators and a mean dry biomass of 164 g m$^{-2}$ (*Fowler et al., 2013*). In New Zealand, where it is well established in many lowland forests remnants, *T. fluminensis* had a mean biomass of 455 g m$^{-2}$ where present and biomasses of up to 1400 g m$^{-2}$ have been observed (*Kelly & Skipworth, 1984a*). At these higher biomass levels, *T. fluminensis* forms a dense ground cover, up to 60 cm deep, which inhibits the growth of native seedlings, preventing native forest from regenerating with a corresponding decline in invertebrate density (*Toft, Harris & Williams, 2001*) and alteration of the nutrient availability (*Standish et al., 2004*) and soil microfauna (*Yeates & Williams, 2001*). In New Zealand, *T. fluminensis* spreads exclusively by vegetative means. A stem of *T. fluminensis* is easily broken and a segment of plant over 1 cm in length and containing a node has a high probability of survival and regrowth (*Kelly & Skipworth, 1984a*). Its prolific growth capacity, lack of

Corresponding author
Alex James,
alex.james@canterbury.ac.nz

natural predators and ability to stifle the regenerating native plants growth in lowland forest remnants in particular has led to *T. fluminensis* being included on New Zealand's National Pest Plant Accord (*2013*), banning it from sale, propagation and distribution.

The presence of *T. fluminensis* is strongly correlated with anthropogenic variables (*Butcher & Kelly, 2011*). It is also known to be light-sensitive: the standing crop of *T. fluminensis* is strongly positively correlated with light intensity, with the highest densities being found at light levels of up to 30–50% of open ground. For light levels above this, the standing crop is still high but not related to light intensity (*Maule et al., 1995*). *Standish, Robertson & Williams (2001)* showed that native seedling richness and abundance decreased exponentially in plots with increasing Tradescantia biomass. A similar effect was reported by *McAlpine, Lamoureaux & Westbrooke (2015)*. This is thought to be primarily due to the decreased light levels that *T. fluminensis* causes: below a mat of 100% *T. fluminensis* cover, light level was reduced to less than 1%. Newly germinated seed are found underneath *T. fluminensis* ground coverage but often these do not develop into seedlings (*Kelly & Skipworth, 1984a*; *Standish, Robertson & Williams, 2001*).

Until 2012, control of *T. fluminensis* in New Zealand has been either by herbicide application or hand weeding. Neither of these methods can achieve complete removal and prevent regrowth (*Kelly & Skipworth, 1984b*), so artificial shading has been suggested as another possible control method (*Standish, 2002*). Shading is not intended to remove the plant entirely but to reduce the biomass to a level at which native woody plants can overtop and outcompete *T. fluminensis*. More recently, biological control methods including three species of criocerine beetles and a fungus have been tested for release in New Zealand (*Fowler et al., 2013*).

In this work, we will present data for the growth of individual *T. fluminensis* plants in both low and high light conditions. We use summary statistics derived from these data to parameterise a stochastic, branching-process model, which is then used to predict the overall change in the biomass of a plant under different light conditions. We then compare the model predictions to different summary statistics to test the validity of the modelling assumptions. The results are used to draw conclusions about the underlying mechanisms for growth in an individual plant.

## METHODS

### Field experiments

Two experimental sites were established on the North and South Islands of New Zealand. The South Island site, "Conway Flat" (43°S, 173°E, 50 m altitude), was established in late February 2007 and located in a 10.5 ha parcel of remnant podocarp-hardwood forest along a river flat, which had undergone a large flood in March 1988. To exclude large herbivores, mainly deer, temporary electric fences were erected at the beginning of the study. However, several months after commencement of the study, feral pigs entered the plots so rabbit fencing (1 m in height, 50 mm mesh size) was fixed around the bottom of the electric fence. The North Island site was established in June 2007 and was located in a 2 ha area of remnant podocarp-hardwood forest on a river floodplain southwest of Hamilton at Te

Pahu (37.5°S, 175°E, 40 m altitude). Herbivory was not a problem at this site so fencing was not required.

At each site, five blocks (replicates) were selected in areas having similar cover of *T. fluminensis* with each block having regions of both high and low light levels (canopy edge and under canopy respectively). One plot per block was established in each of the high and low light areas giving 10 plots (5 high light and 5 low light) at each site. At Conway Flat, 12 *T. fluminensis* plants per plot were randomly selected by searching for the base ends of the plant. Only plants that had a minimum of five green leaves on five nodes plus a base end with roots were chosen; these were defined as "complete plants." The total number of complete plants at this site was 120 (12 plants × 2 light levels × 5 replicates). At Hamilton, the mats of *T. fluminensis* were denser than those at Conway Flat, making it difficult to locate the base of a plant without damaging it and causing major disturbance to the surrounding sward. To avoid this, a combination of complete plants and partial plants was selected. Complete plants were selected as for the Conway site, except that only 5 plants were chosen from both high and low light plots in only two blocks, giving a total of 20 complete plants. In addition, 120 aerial tips, i.e., tips visible at the top of the plant mat, of plants were chosen (defined as "partial plants"): 5 replicates of 12 plants per plot at both light levels. To distinguish chosen plants from others in the sward, a jewellery tag was placed several nodes forward from the basal end (tag 1) on complete plants and on the internode immediately below the oldest leaf-bearing node (tag 2) on both complete and partial plants.

Data collection at both sites occurred once every 3 months for a 12 month period (late February 2007 to early March 2008 for Conway and June 2007 to June 2008 for Hamilton). For complete plants, during the first three censuses, a drawing was made for each plant noting which nodes had leaves present and the occurrence of any branching or flowering. The length between each successive node (i.e., the internode length) from the basal end to tag 1 was measured, as were the internode lengths nodes from tag 2 forward to the tip of the plant. Internode lengths at the tip were measured to the last measurable node (defined as $\geq 0.5$ cm in length). Separate counts were made for mature (those expanded to $\geq 3$ cm) and immature ($<3$ cm) leaves. When branching occurred, a wire tag was placed at the node where it joined the main stem and classified as either a green shoot ($<1$ cm long), a short branch (1–10 cm long), or a green branch ($\geq 10$ cm long). On the fourth and fifth censuses, full measurements were taken only for two plants per plot and the main stem tips of the remaining 10 plants were tagged only. Where necessary, measurements were taken of any loss of stem length at bases and bases retagged. After the final (i.e., fifth) census, plants were harvested one block at a time. Individual plants were carefully pulled from the ground and placed in large numbered plastic bags for measurement at the lab. Before measurement, the roots of each plant were washed and surface water removed with a paper towel. All internode lengths were measured and the root position for the main stem, the existence of flower buds and the positions of branches were noted. The number of nodes was counted for individual branches. Leaf area, fresh weight and dry weight were recorded for the main stem section of each plant. The combined leaf area, fresh weight and dry weight of all branches were also recorded. A LiCor leaf area meter was used to measure leaf areas. Plants

were split up into smaller lots when necessary and dried in brown paper bags at 65–70 °C for at least 4 days to measure dry weights.

Partial plants were measured identically to the complete plants except the measurements were only for the leaf-bearing stem forward to the tip. Because plants were growing quickly, at subsequent visits it became necessary to place extra tags on the internode adjacent to the last measurable tip. Notes were made on the plant drawing of the number of internodes between these tags to avoid problems should the tags slip (e.g. through the death of a leaf).

To summarise, data were collected from four treatments: the Conway Flat site under high and low light and the Hamilton site under high and low light. In each environment, 60 plants were measured at five time points over the course of one year period giving 300 plant measurements at each of the four treatments.

## Model parameter estimates

From the data collected, we calculated the following quantities for each of the four environments.

1. **Tip growth rate** ($\lambda_G$): the mean rate of adding new nodes for a single stem (nodes stem$^{-1}$ year$^{-1}$)

   This is the number of nodes that was added to the front of the main stem between the first and final censuses, averaged across the 60 plants measured.

2. **Death rate** ($\lambda_D$): the mean death rate of nodes at the base of a stem (nodes base$^{-1}$ year$^{-1}$).

   This is the number of nodes which were lost from the rear of the main stem between the first and final censuses.

3. **Branching age** ($A$): the mean age of a node when it branches (years).

   For each node that branched during the study period, the time between the node first appearing and the subsequent side stem appearing was calculated. As censuses were taken at three-monthly intervals, this is an approximate value. Nodes that were already present at the initial census and branched subsequently were excluded as the ages of these parent nodes are unknown.

4. **Branching probability** ($P$): the probability of a node branching.

   The total number of main stem nodes across all plants that produced a side stem was divided by the total number of main stem nodes. Nodes and side stems that were already present at the initial census were excluded as the age of the side stem was unknown. The analysis of branching ages described above revealed that a negligible number of nodes branched at an age greater than 6 months. Nodes that appeared less than 6 months prior to the final census were, therefore, also excluded as they may have branched subsequently to the final census. Hence, only nodes that appeared between the initial and the third census were used in this calculation. All side stems that originated from one of these nodes at any census were included.

5. **Total growth rate** ($\lambda_T$): the net rate of increase in the number of nodes per plant (nodes year$^{-1}$).

**Table 1 Data summary statistics.** Summary statistics calculated from the data collected at two sites under two light conditions. Each cell shows: mean ± standard deviation (95% confidence interval about the mean).

| Treatment | $\lambda_G$, tip growth rate (nodes yr$^{-1}$) | $\lambda_D$, basal death rate (nodes yr$^{-1}$) | A, mean age at branching (yr) | P, Pr (node branching) | $\lambda_T$, net rate of increase in number of nodes (nodes yr$^{-1}$) | B, rate of increase in number of stems (stems yr$^{-1}$) |
|---|---|---|---|---|---|---|
| Conway high light | 6.5 ± 5.6 | 2.1 ± 2.8 | 0.38 ± 0.13 | 0.042 | 5.3 ± 6.3 | 0.26 ± 0.58 |
| (N = 46) | (5.0, 8.1) | (1.4, 3.0) | (0.30, 0.45) | (0.022, 0.071) | (3.4, 7.1) | (0.11, 0.46) |
| Conway low light | 6.0 ± 4.5 | 5.9 ± 6.5 | 0.36 ± 0.13 | 0.030 | 0.12 ± 8.2 | 0.17 ± 0.38 |
| (N = 46) | (4.8, 7.3) | (4.1, 7.8) | (0.28, 0.46) | (0.012, 0.060) | (−2.2, 2.5) | (0.065, 0.28) |
| Hamilton high light | 9.3 ± 8.2 | 2.9 ± 5.6 | 0.14 ± 0.23 | 0.023 | 9.8 ± 9.6 | 0.27 ± 0.48 |
| (N = 70) | (7.5, 11.3) | (1.7, 4.3) | (0.06, 0.25) | (0.014, 0.036) | (7.6, 12.0) | (0.17, 0.39) |
| Hamilton low light | 9.8 ± 5.1 | 2.1 ± 5.0 | 0.38 ± 0.18 | 0.004 | 9.0 ± 8.4 | 0.03 ± 0.17 |
| (N = 70) | (8.6, 11.1) | (0.99, 3.2) | (0.25, 0.5) | (0.001, 0.014) | (7.0, 11.0) | (0.00, 0.071) |

This is the net change in the total number of nodes between the initial and final census, averaged across all plants in each treatment. The net change consists of the sum of nodes that were added to the front of the main stem and side stems minus nodes lost from the rear of the plant.

6. **Branching rate** (B): the rate of increase in the number of stems per plant (stems year$^{-1}$). This was the average number of new stems (including sub-branches) that were added to the plant during the year.

Each of the continuous variables was tested for differences between sites using a two-tailed t-test. The probability variable was tested for differences using a two-sided binomial exact test. All variables were then tested for differences between high and low light environments. For variables with significant between-site differences, this was done separately for each site; for variables with no significant between-site differences, the data were aggregated across the two sites.

## RESULTS

### Field experiment

Table 1 gives the expected value, standard deviation and 95% confidence intervals (found by boot-strapping for non-normal data) for each variable for the four different treatments. Note that P, the probability of a node branching is calculated from the data as a whole, rather than as a mean of each plant and therefore does not have a standard deviation. The confidence interval in this case is found from the binomial distribution. Of the six variables examined, five were significantly different between sites ($\lambda_G, p < 10^{-4}; \lambda_D, p < 0.05; A, p < 0.01; \lambda_T, p < 10^{-4}$). The branching rate, B, showed no significant difference ($p = 0.24$) and the branching probability, P, showed a close to significant difference ($p < 0.1$) between the two sites.

Data from the Hamilton site showed no significant difference between the high and low light environments for four variables ($\lambda_G, p = 0.67; \lambda_D, p = 0.38; A, p = 0.18; \lambda_T, p = 0.59$), but there was a small difference in the branching probability ($P, p < 0.1$). At the Conway

site, there was a significant difference between high and low light in the total growth rate ($\lambda_T, p < 0.005$) and the death rate ($\lambda_D, p < 0.001$) but no significant difference in the tip growth rate ($\lambda_G, p < 0.1$), the branching age ($A, p > 0.1$) or the branching probability ($P, p > 0.1$).

Figure 1(A, D, G, J) shows the time series for the total number of nodes on each plant (grey curves) and averaged across all plants in that treatment (black curve). Clearly, there was a wide range of growth rates for different plants in the same environment. Figure 1(B, E, H, K) shows the frequency distribution of the total growth rate, $\lambda_T$, for plants in each environment. In all four environments, some plants had a net decrease in number of nodes during the measurement period: this was due to the loss of nodes at the base of the plant (basal death) outstripping the increase of nodes at the tips. Figure 1(C, F, I, L) shows the distribution of the rate of increase in number of stems, $B$. In all environments, the majority of plants ($>70\%$) did not branch during the measurement period, giving $B = 0$ for those plants.

## Individual plant stochastic growth model

Rather than following each node and the relationships among nodes individually, the model describes the plant in terms of four key variables:

- The number of young nodes, $N_Y$. Young nodes have the potential to branch and create a new side stem.
- The number of old nodes, $N_O$. Old nodes no longer have the potential to branch, either because they are too old or because they have already branched previously. (Data indicate that it is extremely rare for a single node to be the parent for more than one side stem.)
- The number of tip nodes, $T$.
- The number of nodes (young or old) which have no live parent, i.e., basal nodes, $N_B$. Basal nodes are able to die. Note that basal nodes are already accounted for in the total number of nodes.

The total number of live nodes is $N_Y + N_O + T$. The basal nodes category and the total live nodes are not mutually exclusive: basal nodes are a subset of total live nodes. The model makes no distinction between the main stem and side stems, i.e., nodes on a side stem behave in the same fashion as nodes on the main stem.

The model is based on Poisson processes representing four types of event:

- **Tip growth.** Any stem on the plant can add an additional node at the tip. The tip growth rate is denoted $\lambda_G$ per tip per unit time. The effect of this event is to increase the number of young nodes by 1 (there is no net change in the number of tips). Tip nodes are not able to produce side stems.
- **Node branching.** A young node can branch, creating a new side stem consisting of a single new tip. The node branching rate is denoted $\lambda_B$ per young node per unit time. Since the parent node cannot branch again subsequently, it transitions from being a young node to an old node. The effect of this event is therefore to increase the number of

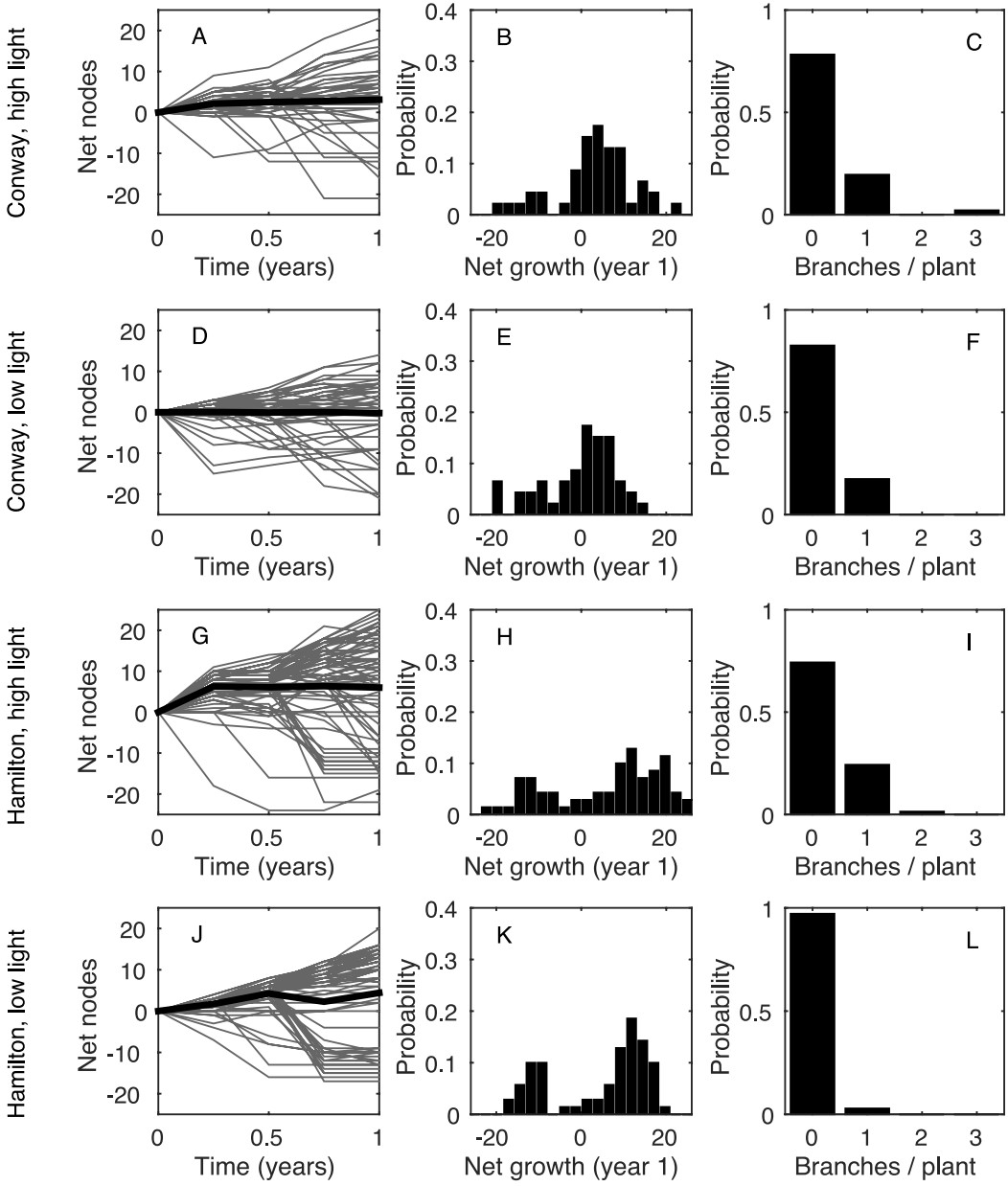

**Figure 1  Data summary.** (A, D, G, J): net change in the number of nodes from the first census (each grey line shows an individual plant; black line shows the mean). (B, E, H, K): frequency distribution of the net change in nodes between the first and last censuses. (C, F, I, L): frequency distribution of the increase in the number of stems between the first and last censuses. Each row shows a different environment (A, B, C—Conway high light; D, E, F—Conway low light; G, H, I—Hamilton high light; J, K, L—Hamilton low light).

tips by 1, to increase the number of old nodes by 1 and to decrease the number of young nodes by 1.

- **Node ageing.** A young node can transition into the old node category (and cannot subsequently create new side stems), representing a natural ageing process. The ageing rate is denoted $\lambda_A$ per young node per unit time.
**Table 2 Poisson process summary.** Summary of the four Poisson processes used in the stochastic model.

| Event type | Poisson event rate | Effect on state variables |
|---|---|---|
| Tip growth | $\lambda_G T$ | $N_Y \to N_Y + 1$ |
| Node branching | $\lambda_B N_Y$ | $T \to T + 1; N_O \to N_O + 1; N_Y \to N_Y - 1$ |
| Node ageing | $\lambda_A N_Y$ | $N_Y \to N_Y - 1; N_O \to N_O + 1$ |
| Basal node death | $\lambda_D \lambda_A N_B / (\lambda_A + \lambda_B)$ | $N_O \to N_O - 1$ |
| Basal node death exposing an additional basal node | $\lambda_D \lambda_B N_B / (\lambda_A + \lambda_B)$ | $N_O \to N_O - 1; N_B \to N_B + 1$ |

- **Basal node death.** The basal node death rate is denoted $\lambda_D$ per basal node per unit time. Basal node death decreases the number of old nodes by 1 and exposes either 1 new basal node (if the dead node had not branched) or 2 new basal nodes (if the dead node had branched); hence the net effect is either to leave the number of basal nodes unchanged or to increase the number of basal nodes by 1. In the long term, the proportion of nodes that branch before transitioning into the old category is $\frac{\lambda_B}{\lambda_A + \lambda_B}$. Hence, this is taken as the probability that this event increases the number of basal nodes by 1. (Implicit in this is the assumption that the main stem is sufficiently long that all nodes have transitioned into the old node category before they become basal nodes.)

Figure 2 shows a schematic to illustrate the four types of event and Table 2 summarises their rates and effects. The model has no explicit spatial structure because individual nodes are not tracked; only the total number of each type of node is known. However, once these numbers are known, an example spatial structure can be inferred. The assumption that each event type is a Poisson process means that events occur non-simultaneously and the process is memoryless (i.e., depends only on the current state of the plant, not on its prior history). Each simulation of this stochastic model gives a different realisation of an individual plant. By doing repeated simulations, summary statistics, such as the mean rate of increase in the total number of nodes, can be calculated.

The four parameters of the stochastic model can be estimated from the data. The tip growth rate $\lambda_G$ and the basal node death rate, $\lambda_D$, are taken directly from Table 1. By considering the expected time at which a node branches conditional on it not having transitioned from young to old before the branching event occurs, the mean age of a parent node when it branches is

$$A = \int_0^\infty \lambda_A e^{-\lambda_A t'} \int_0^{t'} t \, \lambda_B e^{-\lambda_B t} \, dt \, dt' = \frac{\lambda_B}{(\lambda_B + \lambda_A)^2}.$$

The probability of a node branching is

$$P = \frac{\lambda_B}{\lambda_B + \lambda_A}.$$

Taking the values of $A$ and $P$ from Table 1, these two equations can be solved simultaneously to give values for $\lambda_B$ and $\lambda_A$. The initial conditions for the model were chosen to match the state of the plants at the first field experiment census. The initial total number

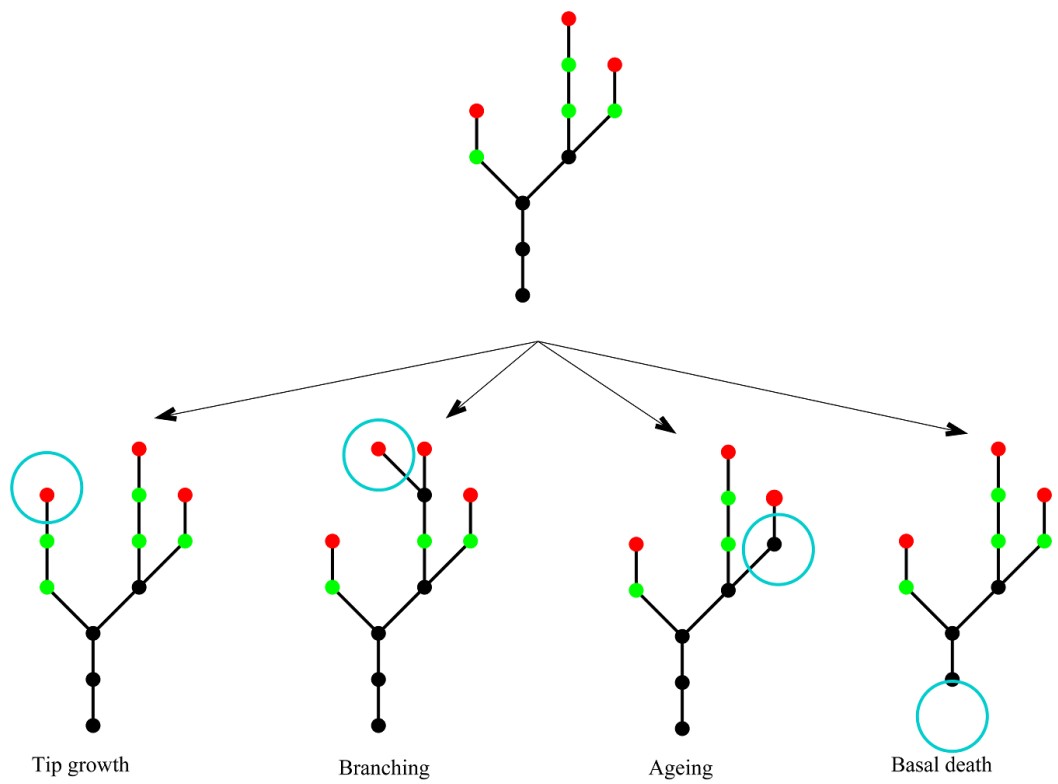

**Figure 2 Model schematic.** A schematic showing the four different events in the model. Young nodes (i.e. nodes with the potential to branch) are shown in green, old nodes (i.e. nodes without the potential to branch) are shown in black, tip nodes are shown in red. The four event types are: **tip growth**—a new young node is created at a tip; **branching**—a young node branches creating a new side stem; **node ageing**—a young node becomes an old node; **basal node death**—an old basal node dies.

of nodes ($N_Y + N_O + T$) was the mean total number of nodes at the first census. Over a large number of simulations, the initial number of tips was given the same probability distribution as the number of tips on plants at the first census. As it is not possible to distinguish between young and old nodes in the data, the initial number of young nodes was estimated from the model prediction for the mean number of young nodes per stem, $\frac{\lambda_G}{\lambda_B + \lambda_A}$. Hence, each plant was initialised with $N_Y = T\frac{\lambda_G}{\lambda_B + \lambda_A}$, though changing this initial condition has very little effect on the model output. Since none of the event rates depend on $N_O$, the model is insensitive to the initial number of old nodes on the plant. Each plant was assumed to have one basal node initially.

## Deterministic growth model

For a simple Poisson process of this type, the average behaviour of the stochastic simulations can be described by differential equations. For this, process the mean-field differential equations are given by

$$\frac{dT}{dt} = \lambda_B N_Y,$$
$$\frac{dN_Y}{dt} = \lambda_G T - (\lambda_A + \lambda_B)N_Y,$$

$$\frac{dN_o}{dt} = (\lambda_A + \lambda_B)N_Y - \lambda_D N_B,$$
$$\frac{dN_B}{dt} = \frac{\lambda_D \lambda_B}{\lambda_A + \lambda_B} N_B.$$

Note that the number of basal nodes is independent of the total number of live nodes and can be solved separately to give exponential growth of basal nodes over time at rate

$$\frac{\lambda_D \lambda_B}{\lambda_A + \lambda_B}.$$

Similarly, the equations for tips and young nodes are uncoupled from the rest of the system and can be solved to show that the number of tips and young nodes will increase exponentially over time at rate

$$\frac{-(\lambda_A + \lambda_B) + \sqrt{(\lambda_A + \lambda_B)^2 + 4\lambda_G \lambda_B}}{2}.$$

Thus, the overall behaviour of system is given by the behaviour of the old nodes. If these are being lost by basal death faster than they are increasing by branching and aging of young nodes, then the plant will die in the long term (i.e., $N_Y + N_O + T$ will eventually reach zero). The condition for the death of a plant is therefore

$$\frac{\lambda_D \lambda_B}{\lambda_A + \lambda_B} > \frac{-(\lambda_A + \lambda_B) + \sqrt{(\lambda_A + \lambda_B)^2 + 4\lambda_G \lambda_B}}{2}.$$

## Model validation

The model, in either its stochastic or mean field format, only requires the first four statistics shown in Table 1 as inputs. This allows us to use the model to predict the final two statistics (total growth rate, $\lambda_T$, and branching rate, $B$) and compare these predictions with the data. The predictions are made using the behaviour of a large number of repeat simulations of the stochastic model or, in the case of the mean total growth rate, using the solution of the mean-field deterministic model. Figure 3 shows the same results as in Fig. 1, but using the model predictions rather than the observed data; the corresponding summary statistics are shown in Fig. 4 and Table 3. In all four environments, model predictions for the mean total growth rate and the mean branching rate lie within the 95% confidence intervals estimated from the data (Fig. 4).

## Variance

Although the above simple Poisson process model predicts the mean plant characteristics with reasonable accuracy there are some differences in the distributions of these quantities (compare central column in Figs. 1 and 3). In particular, the variance in the net growth rate is higher in the data than in the stochastic model. Table 1 shows that, of the four model inputs, main stem growth rate $\lambda_G$ and basal death rate $\lambda_D$ vary widely between plants in the same treatment. The basal death rate has a coefficient of variation (CV) greater than one

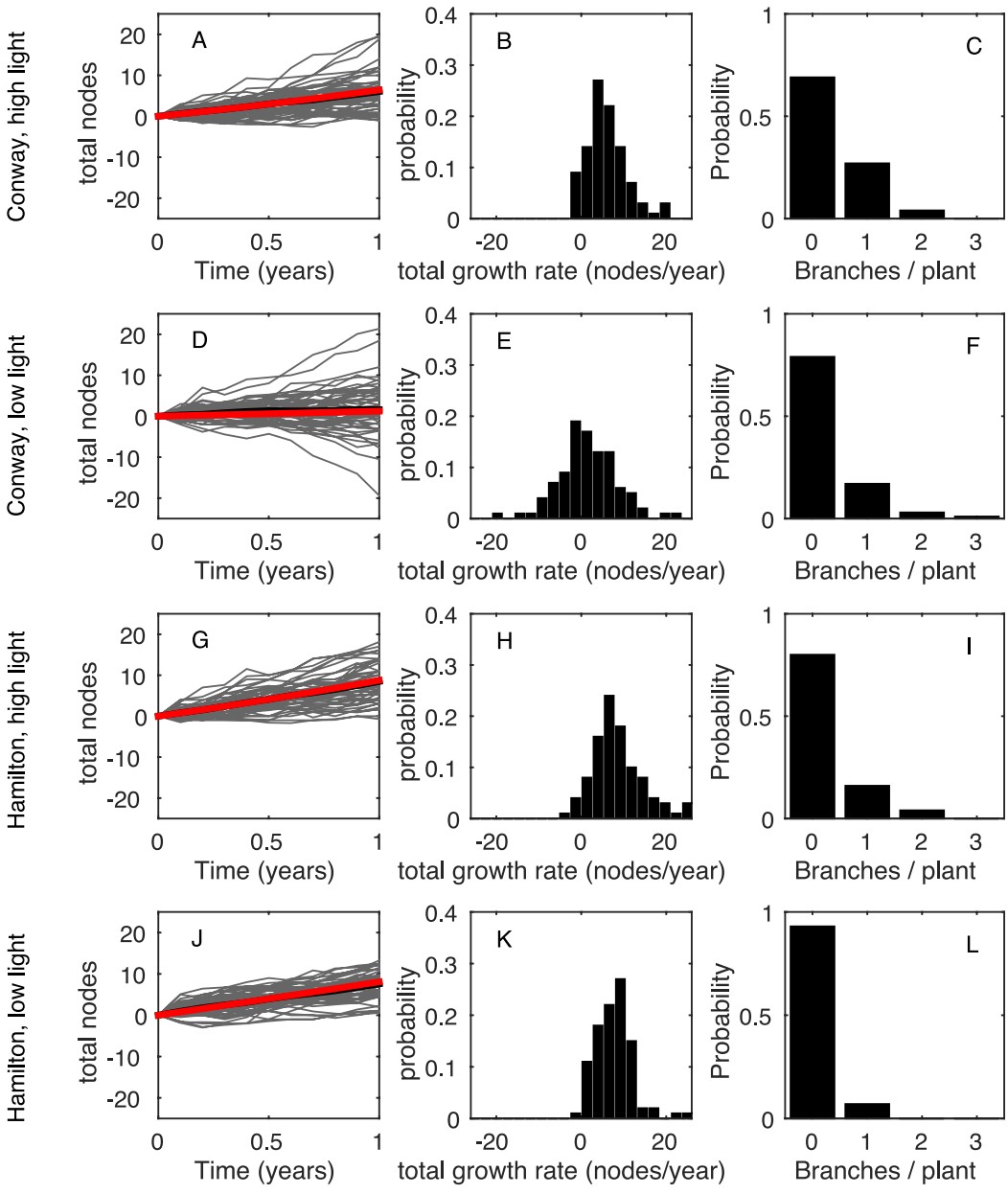

**Figure 3  Poisson model results.** The same results as Fig. 1 but for 100 simulations of the model using the parameter values given in Table 1. The expected net increase in total nodes is given as the mean of the stochastic process (black line) and as the deterministic prediction (red line). If the number of simulations is increased these two lines become indistinguishable.

and the growth rate has CV $\approx$ 1 for all four cases. Conversely, branching age $A$ has CV less than 1 for three of the four treatments.

Table 4 shows the variation seen in the model outputs and the data. The simple Poisson process model described above fails to capture the variance in the total growth rate, though it makes a reasonable estimate for the branching rate. As the underlying plant model is based on a stochastic process, additional sources of variance can be easily included. As the

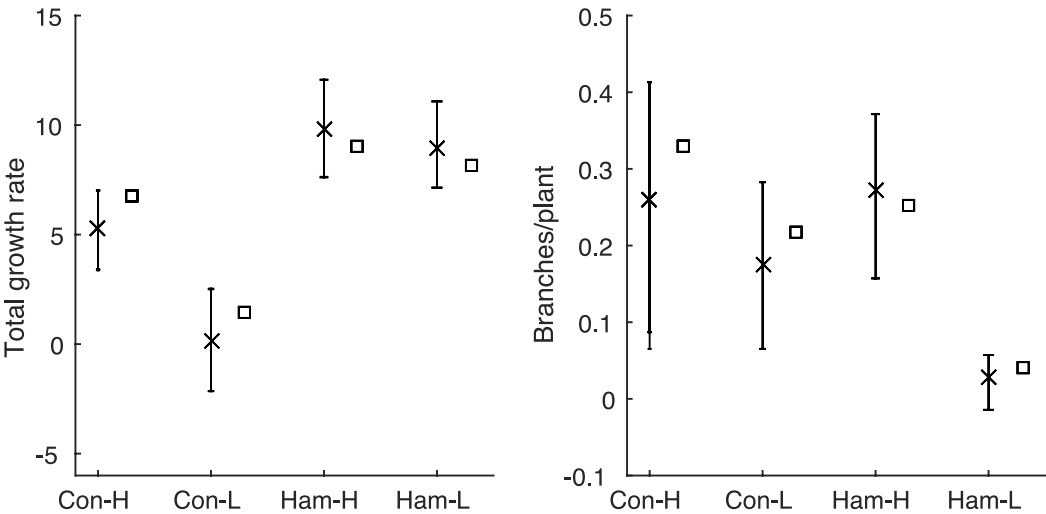

**Figure 4 Comparing Poisson model and data.** The summary statistics and the values predicted by the model for each environment: mean values calculated from the data are shown by crosses with 95% confidence intervals; mean values predicted by the model are shown as squares. The model prediction lies within the 95% confidence interval in all cases.

**Table 3 Model predictions.** Model predictions for the net rate of increase, the total number of nodes and the branching rate in four environments. All values are within the 95% confidence intervals calculated from the data.

| Treatment | $\lambda_T$, net rate of increase in total number of node (nodes yr$^{-1}$) | $B$, rate of increase in number of stems (stems yr$^{-1}$) |
|---|---|---|
| Conway high light | 6.8 | 0.33 |
| Conway low light | 1.5 | 0.22 |
| Hamilton high light | 9.0 | 0.25 |
| Hamilton low light | 8.1 | 0.039 |

**Table 4 Model predictions of variation.** Data and model predictions of the coefficient of variation for the model output parameters. The simple Poisson model often fails to capture the variance in the total growth rate, whereas the high-variance model is more accurate in this respect.

| Treatment | $CV(\lambda_T)$, net rate of increase in total number of node (nodes yr$^{-1}$) | | | $CV(B)$, rate of increase in number of stems (stems yr$^{-1}$) | | |
|---|---|---|---|---|---|---|
| | Data | Poisson model | High-variance model | Data | Poisson model | High-variance model |
| Conway high light | 1.21 | 0.76 | 1.28 | 2.20 | 1.79 | 1.86 |
| Conway low light | 66.38 | 4.02 | 6.12 | 2.20 | 2.21 | 2.23 |
| Hamilton high light | 0.98 | 0.67 | 1.15 | 1.77 | 2.10 | 2.13 |
| Hamilton low light | 0.94 | 0.47 | 0.92 | 5.87 | 5.02 | 5.27 |

branching process appears to be adequately captured and the parameters that govern it have relatively low variance among plants, this part of the model was not changed. The high-variance model focuses on capturing the variance in the total growth rate by altering the main stem growth rate $\lambda_G$ and basal death rate $\lambda_D$.

The main stem growth rate $\lambda_G$, is approximately normally distributed across plants at two of the four treatments (Lilliefors test: Con high $p = 0.001$, Con low $p = 0.16$, Ham high $p = 0.21$, Ham low $p = 0.001$). Interestingly, it has approximately the same coefficient of variation at each treatment, though the mean varies widely. The basal death rate $\lambda_D$ is geometrically distributed at three of the four treatments ($\chi^2$ goodness of fit test: Con high $p = 0.004$, Con low $p = 0.82$, Ham high $p = 0.08$, Ham low $p = 0.23$). To capture this additional variance we rerun the model but, in each realisation, generate new values for the growth and death rate parameters. We assume the growth parameter is normally distributed with mean $\lambda_G$ for that treatment and coefficient of variation 0.8

$$\lambda_{G_i} \sim N\left(\lambda_G, (0.8\lambda_G)^2\right).$$

In the data the death rate of an individual plant is discrete and well-approximated by a geometric distribution. In the model the death rate is a continuous variable so we assume it is exponentially distributed with mean $\lambda_D$ for that treatment

$$\lambda_D \sim Exp(\lambda_D).$$

These additional sources of among-plant variability increase the variance of the model realisations (Fig. 5). The coefficient of variation of the model outputs for each treatment is shown in Table 4. This high-variance model is a better fit to the data than the original model, although it still fails in the case of the Conway low light treatment, which has an extremely high CV (66) in the total growth $\lambda_T$.

## Probability of ultimate extinction

Using the deterministic model, we have calculated the conditions under which a plant is guaranteed to go extinct. Using a stochastic model allows this idea to be extended to predict the probability a plant will go extinct in cases where the criterion derived in the deterministic case does not hold. To assess the chance of survival for a plant at the different treatment levels, we calculate the probability of ultimate extinction (PUE). This is defined as the fraction of realisations (i.e., individual plants in the data or realisations of the stochastic process) that undergo a decrease in the total number of nodes after one year, i.e., $\lambda_T < 0$. Table 5 shows PUE for the data with the 95% confidence interval and the predictions from each model. The original Poisson model fails to capture PUE for any of the treatments whereas the high-variance model prediction falls within the confidence intervals in all four cases.

## DISCUSSION

We have formulated a mathematical model for the growth of individual *T. fluminensis* plants using empirical data from plants at two geographic sites in New Zealand and at two different light levels for each site. Each plant was measured at five time points, allowing us to infer average rates for key processes such as tip growth, branching and node death. Parameters for the mathematical model were estimated for each environment using four observed quantities: (i) the tip growth rate; (ii) the basal node death rate; (iii)
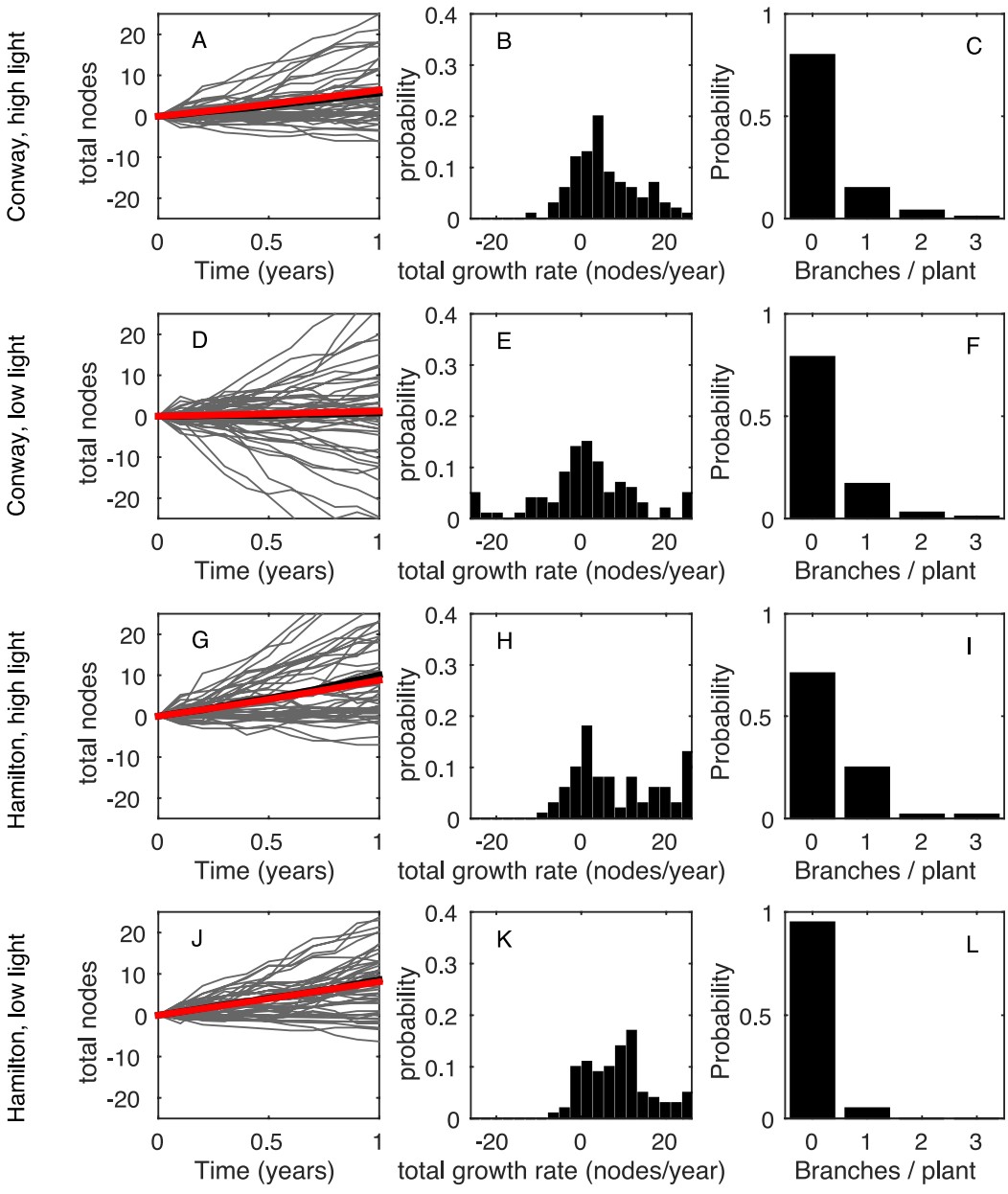

**Figure 5 High variance model results.** The same results as Fig. 3 using the high-variance model. There is very little change in the branching results but the total number of nodes after one year shows a much higher variance. The differential equation (red line) is still a good approximation to the mean of the process.

the proportion of nodes that branch; and (iv) the mean age of those nodes at the time of branching. The model was then used to predict two other observed quantities: the overall growth rate and the branching rate. In all cases, the model predictions for these quantities lay within the 95% confidence intervals given by the data.

The quantities used to parameterise the model came from the same data sets as those that the model was used to predict. This does not, therefore, give a completely independent

**Table 5 Model PUE predictions.** Data and model predictions of the probability of ultimate extinction (PUE). The high-variance model predicts PUE within the 95% confidence intervals of the model for all four treatments. The original model fails to do this in all four cases.

| Treatment | PUE, $P(\lambda_T < 0)$ | | |
|---|---|---|---|
| | Data (95% CI) | Poisson model | High-variance model |
| Conway, high light | 0.17 (0.065, 0.28) | 0.01 | 0.17 |
| Conway, low light | 0.35 (0.21, 0.48) | 0.53 | 0.44 |
| Hamilton, high light | 0.14 (0.071, 0.23) | 0.00 | 0.14 |
| Hamilton, low light | 0.10 (0.043, 0.17) | 0.01 | 0.17 |

model validation. Nevertheless, it does demonstrate that the model is consistent with the observed data and gives some confidence in the model's assumptions. In particular, the model assumed that side stems behave in a manner identical to the main stem and we have found no evidence to contradict this (*Molloy, 2010*). This is useful when considering potential methods of controlling *T. fluminensis*.

The largest discrepancy between the Poisson model and the data is in the variance of the net growth rate between individual plants: real plants exhibit a greater variance than predicted by the stochastic model. The initial model assumes that growth, branching and death events occur as Poisson processes. This accounts for random variability in the times at which these events take place. However, it does not account for any between-plant environmental variability in, for example, microclimate, soil quality or herbivory. In a natural environment, it is likely that plants are subject to substantial environmental variability and the model does not currently account for this. The high-variance model accounts for this environmental variation by allowing individualised growth and death rates for each plant. This additional variance alone appears to improve the model predictions, in particular predictions of the probability of extinction, which rely on a good estimate of among-plant variance. This shows that environmental variability may have little effect on the plant branching rate but does affect the likelihood of growth and death. For example, the high coefficient of variation of these parameters could be explained by herbivory, flood or branch fall events affecting the loss and gain of nodes.

Under both models, a plant will either die in the long term or will undergo sustained growth. For a given set of parameters, the model can be used to predict the proportion of plants that will eventually die. This has obvious applications to control of *T. fluminensis*. The model could be used to test the efficacy of potential control measures, for example shading, manual removal or herbivory. In each case, the model can predict the level of control needed to achieve the death of an individual plant. This will enable competing potential control strategies to be cheaply compared prior to field testing. The treatment with the highest probability of ultimate extinction was the Conway site with low light. The key contributor to this effect was the high rate of basal death, particularly in comparison to the growth rate. In contrast, the Hamilton site with low light had a low branching rate but its high growth rate and low death rate resulted in a very low probability of extinction.

This suggests that control measures that promote node death at the base of the plant or restrict the main stem growth rate will be more effective than those that reduce the number of branching events.

Previous work has shown a difference in *T. fluminensis* biomass and regeneration between high light and low light environments (*Kelly & Skipworth, 1984a*; *Maule et al., 1995*; *Standish, Robertson & Williams, 2001*). This was not observed in the Hamilton data, though there were significant differences in the Conway data. The data also show that the difference in the branching rate between high and low-light environments was small and not significant. The main difference that we found between *T. fluminensis* growth in high and low light levels is in the basal death rate. This is important because the basal death rate was predicted to be a key determinant of plant survival. This suggests that the efficacy of biocontrol agents, which typically affect the branching or tip growth rate, may be enhanced by combining with shading measures, aimed at increasing the basal death rate.

To summarise, we have presented a simple Poisson process model for the growth of *Tradescantia fluminensis*. The model was parameterised with data derived from field experiments and then verified with independent data. The model gave good predictions for the outputs based on the mean values which showed that the underlying assumptions were sound. However, the model had less predictive power for outputs based on variance showing that some assumptions were lacking. We extended the model to include higher variability between plants and the predictions improved. Finally, we discussed the long term survival of the plant under different control methods and predicted that control measures that promote node death at the base of the plant or restrict the main stem growth rate will be more effective than those that reduce the number of branching events. Now established, this model would form a good base for assessing the efficacy of various forms of biocontrol.

### Funding

Funding for this work was provided in part by the Ministry of Business, Innovation and Employment (previously the Foundation for Research, Science and Technology) under the Beating Weeds (contract C09X0504) and Beating Weeds II (contract C09X0905) programmes and a University of Canterbury Women in Engineering scholarship awarded to A P-S. The funders had no role in study design, data collection and analysis, decision to publish, or preparation of the manuscript.

### Grant Disclosures

The following grant information was disclosed by the authors:
Ministry of Business, Innovation and Employment.
Beating Weeds: C09X0504.
Beating Weeds II: C09X0905.
University of Canterbury Women in Engineering scholarship.

## Competing Interests

Shona L. Lamoureaux and Graeme W. Bourdôt are employees of AgResearch, Lincoln, New Zealand.

## Author Contributions

- Alex James and Michael J. Plank analyzed the data, wrote the paper, prepared figures and/or tables, reviewed drafts of the paper.
- Sue M. Molloy conceived and designed the experiments, performed the experiments.
- Agate Ponder-Sutton analyzed the data, reviewed drafts of the paper.
- Shona L. Lamoureaux, Graeme W. Bourdôt and Dave Kelly conceived and designed the experiments, wrote the paper, reviewed drafts of the paper.

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
