# Peer review of "Modelling Tradescantia fluminensis to assess long term survival"

_PeerJ, doi:10.7717/peerj.1013_

## Round 0.1 · original submission · Major Revisions

As you may in the following section, both reviewers consider that your work fits PeerJ standards and therefore should be considered for publications. However, several issues were raised during the revision process. Please address them carefully before further consideration for publication.

On a personal note, I'd like to strongly suggest to change the title of your work. As you may see on Dr. Myerscough revision, the title is misleading generating the expectation for a model that, apart from being validated using a test set, should be tested on the field as a way to reduce the penetration of this species.

·

Basic reporting

Acceptable

Experimental design

Acceptable

Validity of the findings

Acceptable

Additional comments

This manuscript reports on a joint empirical/theoretical paper on the growth of an invasive plant species in New Zealand. In general, the work is expertly performed. The reported worked included survey of the growth of plants over time in the field under a variety of field conditions and the development of two models. Overall, I think the conclusions are sound as reached using these methods. My specific comments are primarily aimed at refinements and extensions to the analysis as well as possibly more elegant solutions. There is one significant problem that should be addressed in a revised manuscript. Namely, the observed data included a number of trajectories in which the number of nodes actually declines overtime. In contrast, model solutions – shown in figure 3 – do not allow for this behavior at all. This is a qualitative lack of fit between the model and data. The authors should address this in their discussion and seek to provide a model that more accurately reflects this heterogeneity. I think the reason for this difference is that the Poisson model does not allow sufficient heterogeneity in growth. Particularly, histograms in figure 3, column two virtually never have negative growth, whereas figure one column, two shows negative growth to be frequently realized in reality, sometimes with large magnitude.

Please find additional detailed comments below.
In general, I was puzzled why a stochastic branching process model was developed. The authors do not compare the variance in growth in observations and model predictions, nor do they look at the first passage times – for instance time to death of the plant. It would seem that all of the main conclusions could be derived from the mean field model. On the other hand, the model is quite elegant. My proposed solution it's not that the stochastic model be dropped, but that additional features, such as the variance and spread, be added to the analysis.
In the estimation of branching age, nodes that were present at the initial census were dropped. These data could possibly be used by estimating age using event time analysis with left censoring rather than throwing the data away.
It is unclear to me why simulation was used to obtain stochastic model solutions. Since the authors are interested primarily in the mean growth rate I think this could be obtained analytically from the mean matrix.
Model validation was relatively simplistic. (Basically, the authors looked out whether or not the mean total growth rate and branching great predicted by the mortal would lie was in the 95% confidence interval's estimated from the data.) I recognize that there are some tricky things that impinge on validating stochastic models (how does one validate a stochastic prediction with a single realization in nature?). But, it would be useful to have some kind of numerical validation, for instance the correlation between predicted and observed values in individual plant growth over time.
As the authors observe, there is a discrepancy between the model and observations in the variance in growth rate across individual plants. As the authors conclude, this almost certainly comes from their choice of model (Poisson). I wonder why the authors did not study an over-dispersed distribution such as the negative binomial distribution, which also could be used for a branching process, and could be interpreted as a Poisson model with a heterogeneity parameter. In contrast to the authors' claim that accurately parameterizing this type of model would require substantially more data, I think it would involve just one or a few more parameters and that the authors probably already have the data in hand.
In the discussion, I would have liked to see the interpretation pushed just a little further. How much shading would be required to induce the plant into a negative growth regime? How different is this in the presence versus absence of biocontrol agents?
One of the most interesting and empirically amenable results of this work is the derivation of a theoretical condition for plant death death. I would have loved for the authors to explore this in greater detail. Do they have data that might validate this prediction? Can this prediction be investigated experimentally? Do the authors have plans to pursue this particular result further?
In table 3 I would have liked to see a representation of uncertainty in these estimates – possibly as histograms of solutions when the parameter uncertainty is propagated through Monte Carlo simulation. At a minimum, the authors should report standard errors.
In conclusion, I commend the authors for a very fine study and express my thanks for their efforts to understand this important environmental problem.
Sincerely yours,
John Drake

·

Basic reporting

The article satisfies all of PeerJ's standards.

Experimental design

The experiments appear to have been described completely conducted rigorously and to a high technical standard. I am not qualified to comment in detail on the statistical methods used on the data. The modelling is correctly explained and carried out.

Validity of the findings

The authors do not really address the use of the model to examine long-term survival as promised in the title. They use the field data to construct and validate the model but stop there. The model is not employed to investigate survival or, for example, to explore the possible effect of control measures (mentioned in line 306). Once the model has been constructed and validated, this should be straightforward to do and the absence of results from the model (beyond validation) makes the article seem incomplete. As such, it is not actually incorrect but it is distinctly lightweight. But I feel, that unless the authors include extra work which uses the model to predict long-term survival, then the title should be changed so that it is clear that the article only concerns model formulation and validation.

It would also have been interesting to have read some discussion on the fact that the high variance of the experimental results compared to the model results come from plants where the number of nodes is declining over the period of the experiment. This suggests that some individual plants in the field may be experiencing increased basal node death compared to the model plants.

---

## Round 0.2 · accepted · Accept

Thanks a lot to the authors of this manuscript by their careful addressing of the issues raised by the reviewers. From my perspective, the paper has been substantially improved compared to the previous version and now is acceptable for publication. Congratulations!